# The Comparisons of Physical Functional Performances between Older Adults with and without Regular Physical Activity in Two Different Living Settings

**DOI:** 10.3390/ijerph18073561

**Published:** 2021-03-30

**Authors:** I-Fang Cheng, Li-Chieh Kuo, Yi-Jung Tsai, Fong-Chin Su

**Affiliations:** 1Department of Biomedical Engineering, National Cheng Kung University, Tainan 701, Taiwan; z9704007@ncku.edu.tw; 2Department of Occupational Therapy, National Cheng Kung University, Tainan 701, Taiwan; jkkuo@mail.ncku.edu.tw; 3Department of Medical Research, E-Da Hospital, Kaohsiung 824, Taiwan; ed108805@edah.org.tw; 4Medical College, I-Shou University, Kaohsiung 840, Taiwan; 5Medical Device Innovation Center, National Cheng Kung University, Tainan 701, Taiwan

**Keywords:** healthy aging, regular physical activity, physical function, community-dwelling, senior day care center

## Abstract

We compared the physical function performances of community-dwelling and day care center older adults with and without regular physical activity (PA). A total of 163 Taiwanese older adults living in rural communities participated. PA habits and physical functional performances were assessed. The participants were divided into community-dwelling (CD) and senior day care (DC) center groups that were further classified into regular physical activity (RPA) and non-physical activity (NPA) subgroups. Comparison took place between subgroups. In the CD group, only the grip strength, pinch strength, and box and blocks test scored significantly better for the participants with regular PA. Muscle strength, flexibility, and three items of functional ability of participants with regular PA were significantly better in the DC group. An active lifestyle contributes to a good old-age life. The effective amount of PA and the reduction of sedentary time should be advocated to prevent frailty and disability in older adults.

## 1. Introduction

Physical activity (PA) means body activity that is carried out by muscle requiring more energy than resting, while exercise (EX) means a planned, structured, repetitive, and intentional activity [1]. Regular PA is defined as habitual PA that is behaving in a regular manner. Regular PA and exercise play important roles in maintaining the quality of health in aging [2], because they could be efficient countermeasures for age-related muscle strength loss [3,4] and disability [5]. For the middle-aged population, regular exercise is effective to maintain muscle strength and physical performance and to prevent sarcopenia in older age [6,7]. The older adults with habitual PA can retain their strength more than the sedentary ones [8]. Older adults who have been habitually active for more than 10 years have lower bone loss and retain better balance than those have been consistently habitually inactive [9]. PA is beneficial for the prevention and the risk reduction of chronic diseases and mental setback among older adults [10,11,12]. The specific list includes all-cause mortality [13], cardiovascular disease [14], type 2 diabetes [15], cancers [16,17], dementia [18,19,20], and depression [21].

Previous studies have shown the positive effects of exercising on muscle strength, endurance [22], and reversing frailty in the elderly [23], and that regular training can improve the functional performance of the elderly [22,24] and, moreover, prevent falls [25,26,27]. According to the Global Health Observatory data [28], insufficient PA in adults in 2016 was 28%, globally. Physical inactivity is especially common in older adults with physical and mental decline. Despite the recommendation that healthy older adults engage in 150 min of moderate or 75 min of vigorous PA per week [29], over half of them fail to meet these guidelines. While inactive or frail older adults should engage in 100 min of moderate or 300 min of light PA per week, few of them achieve this in an assisted living environment [30]. Older adults having no regular PA or longer sitting times are at a higher risk of prefrailty [31]. Too much daily sedentary time can lead to severe frailty with metabolic and musculoskeletal problems [32]. The sedentary behavior (SB) in daily living is associated with adverse heath events in older adults [33].

Sedentary behavior varies with an individual’s physical condition and living environment. Frail older adults using assisted living facilities spend more sedentary time than community-dwelling older adults. They also demonstrate a significantly lower level of functional fitness and inactivity [33]. The aged population is rising rapidly in the world and it is still increasing, with senior day care centers (nonresidential assisted living facilities) becoming a newly booming option for more and more families by providing daytime care services on weekdays. The clients are frail older adults who cannot be safely alone at home when their families are out for work. They are in a stage between living independently and residing in an assisted living facility. They may return to independent living if their functional abilities improve; conversely, they may be sent to resident assisted living institutions if frailty or disabilities worsen. Regular PA could be a countermeasure to revert or prevent frailty in assisted living [34]. A large amount of literature indicates that physically frail older adults can benefit from exercise intervention. Exercise programs have positive effects on ADL (activities of daily living) and IADL (instrumental activities of daily living) when applied for frail older adults [35]. Exercises with low-to-moderate intensity improve muscle strength, endurance, and gait [36], reduce the risk for falling [37], and maintain the physical functions of frail older adults [38,39]. Even for the very frail elderly, balance exercise can improve static balance, while gait exercise can improve dynamic balance and gait functions [40]. However, in most assisted living facilities, sedentary activities such as painting and handcraft making are often included, but physical activities are rare and insufficient.

Effects of physical activity or exercise on healthy older adults are much discussed, however, few studies are concerned with the effects of regular PA in different living settings from viewpoints of physical functional performance, which implies physical capacity to perform a functional activity required in daily life, particularly on the frail ones in assisted living. As to the results of the previous studies mentioned above, regular PA may have different effects on older adults with different health status. The purpose of this study is to compare the physical functional performances of older adults with and without regular PA in two living settings, independent living and day care living groups. We hypothesized that regular physical activity could enhance the physical functional performance of both groups. We also hypothesized that correlations exist between the frequency of physical activity per week and the physical functional performances of community-dwelling older adults.

## 2. Materials and Methods

### 2.1. Subjects

For the purpose of this study, older adults living in rural areas were recruited via two local bureaus and three senior day care centers in Tainan City, Taiwan. All of them were ambulatory with/without assistive devices, and mentally competent to understand instructions. Candidates with major musculoskeletal disease or other disorders affecting their ability to perform the exercises and our physical function tests were not taken in. In total, 163 participants (78.2 ± 7.4 years; 48 males and 115 females) were recruited for this study; all of them met inclusion criteria and were successfully measured. Ethical approval was granted by the National Cheng Kung University Hospital Institutional Review Board (approval number: B-ER-105-126), and written consent was obtained from all participants before the start of the study.

### 2.2. Study Design

The participants were grouped according to their lifestyles: (1) community-dwelling (CD) older adults (healthy individuals who live in a private residence without care support) and (2) physically frail older adults attending senior day care (DC) centers for at least one year. A face-to-face questionnaire was used to survey the PA habits of the participants of the CD group, including type, duration (the length of time for each bout of any specific activity), frequency (the number of times per week they exercise), and intensity (the scores determined by Borg Rating of Perceived Exertion Scale) of the exercises with more accurate screening. Regular PA in this study was defined as engaging in PA at least three times a week for more than 30 min each time. For the participants of the DC group, the PA habits were determined by the daily schedule of activity provided by the day care centers. PA in day care centers was determined as the activities involved in major muscles, for example, walking and Tai Chi rather than static activities such as drawing and handicraft arts. On that basis, each group was further classified into “regular physical activity” (RPA) and “non-physical activity” (NPA) subgroups. The participants who engaged in physical activities less than 30 min for each bout or less than three times a week were classified as NPA subgroup.

The body height was measured using a measuring rod with an accuracy of 0.1 cm. The body weight was measured using a digital medical scale with an accuracy of 0.01 kg. The Body Mass Index (BMI) was calculated by the weight and the height in kg/m^2^. The demographic characteristics and the frequency of physical activities were obtained via questionnaires. Table 1 displays the characteristics of the participants. Muscle strength, flexibility, functional ability, and movement speed were measured by performance-based tests.

### 2.3. Physical Functional Performance

The fitness test for older adults consists of physical and functional components. Physical fitness includes muscle strength, endurance, flexibility, and so forth, and functional fitness is about the ability to perform normal daily activities [41]. According to the previous literature, exercises for healthy older adults have greater influence on physical fitness than on functional status [6,8]. There is not much difference in functional status between them because they are totally independent in activities of daily living (ADL). Therefore, we measured physical functional performance focusing more on physical fitness in the CD group and more on functional fitness in the DC group with respect to frailty, which includes the weakness of grip strength and slow movement [42]. The measured items for the CD group included strength of grip, pinch, knee extensor/flexor, and hip abductor [43]. Box and blocks test (BBT) [44] and chair sit-and-reach test (CSRT) [41,45] were also conducted. For the DC group, we measured grip strength and pinch strength, and conducted BBT and CSRT. Functional fitness was evaluated via sit-to-stand test (STS), modified stepping test (MST) [45], six-meter walking (6-M walk) test [46], and forward reach Test (FRT) [47]. Table 2 contains a brief description of the measured items and the physical functional performance tests. In this study, dominant hand was determined by the hand used the most in daily activities, such as writing and eating, while dominant leg was determined as the leg used to kick a ball.

### 2.4. Statistical Analysis

Mann–Whitney U test was used to compare the variables of physical functional performances between the RPA and NPA of CD and another Mann–Whitney U test was done between RPA and NPA of DC. In the CD group, we used the Pearson correlation coefficient (Pearson’s r) to draw the associations between exercise frequency per week and physical functional performances. In the DC group, the frequency of physical activities was decided by the day care centers, and the whole RPA subgroup engaged in physical activities the same number of times per week, so that it is meaningless to analyze the correlation in this group. The threshold for the statistical significance was defined as *p* = 0.05. All statistical analyses were performed by SPSS version 17.0.

## 3. Results

In the CD group (Table 3), muscle strength and function of upper extremity including dominant grip strengths, bilateral pinch strength, and BBT succeeded significantly better for those with regular PA, but there were no significant differences in lower extremity strength between the older adults with and without regular PA. In the DC group, the older adults with regular PA demonstrated significantly better strength of bilateral grip and dominant pinch. Flexibility of right lower extremity, and functional performances in STS and MST were also significantly better in the older adults with regular PA than without (Table 4). However, walking speed in 6-M walk test and dynamic control in FRT were not significantly different between RPA and NPA subgroups. Figure 1, Figure 2 and Figure 3 show the differences of muscle strength, flexibility, and functional performances between the participants with and without regular PA in both CD and DC groups. Besides, the Pearson correlation analysis revealed no significant correlation between physical activity frequency per week and the physical functional measures in healthy older adults (Figure 4).

## 4. Discussion

The results of this study showed that individuals practicing regular physical PA performed significantly better in several physical functional tasks than their counterparts in both CD and DC group (see Figure 2, Figure 3 and Figure 4). These results confirm those mentioned in the introduction about age-associated loss of muscle strength [3] and muscle strengths in the limbs [4]. We agree that being physically active can prevent or delay the progression of basic ADL disability in aging populations [5].

In this study, there was a difference in physical levels between the two main groups. In the CD group, the RPA subgroup demonstrated greater grip and pinch strength, as well as better functional performance in BBT. In healthy older adults, PA has greater influence on physical fitness—muscle strength and endurance, flexibility, body composition, anaerobic capacity, and aerobic capacity—than on functional status [48]. Since most healthy older adults are totally independent in all self-care, few or no difference exists in assessments of functional ability. Regrettably, expected greater flexibility and strength of major muscle group in the lower extremity for the RPA subgroup did not show in our study. This outcome might be due to insufficient and effective amount of PA. The CD participants may have overestimated their exercise quantity when they reported them. For physically independent older adults, exercise quality not only considers duration and frequency, but also type and intensity. Moderate to vigorous intensity aerobic exercise could be advocated to the physically independent aged population, and muscle-strengthening for major muscle groups should be done more than twice a week to promote their physical fitness. Furthermore, PA on the basis of multiple components is a viable solution for fall prevention among community-dwelling older adults with prefrailty [49]. To avoid getting frail in old age, healthy older adults should engage in vigorous and moderate PA while they are non-frail [50].

Individuals with regular PA in DC demonstrated greater grip and pinch strength, higher flexibility of the lower extremity, and faster movements in BBT, STS, and MST. Subjects had RPA walks faster in the six-meter walking test, but without significant difference. In our study, greater flexibility was demonstrated by the individuals with regular PA in DC centers. Regular PA-induced improvement of flexibility is helpful to reduce falls in older adults because musculoskeletal limitations or deficiencies in the range of motion and the flexibility of the lower extremity may result in loss of balance [51]. Emilio et al. [52] also reported that their proprioception program intervention significantly improved flexibility—which is positively associated with balance control—and reduced the risk of falls in older adults. Regular PA contributes to flexibility improvement significantly. In addition to the effects on physical functions mentioned above, PA also promotes benefits on the cognition of frail older adults [53]. Although PA is an effective intervention to enhance physical capacity such as muscle strength, caution should be taken in type and intensity to optimize functional enhancements in frail older adults [54], since regular PA focused on the specificities of frail older adults can improve their functional levels and reverse the frail status [55]. Surprisingly, no significant correlation was found in this study between the weekly frequency of PA and physical functional performances in the CD group (see Figure 4). Although individuals with regular PA performed better in several tests, exercise frequency was not the most critical part of their physical functional improvements. Zhang et al. stated that even people with less than three exercises a week benefit from the activities if they are active in their daily life [56]. Varying levels of habitual activity have no influence on the musculoskeletal and functional outcome measures either [9]. The total quantity of PA determines the degree of health benefits.

The 2018 Physical Activity Guidelines Advisory Committee Scientific Report recommended that individuals within the public health target gain more benefits by doing moderate-to vigorous PA, while individuals below the target PA range achieve greater benefit by reducing sedentary behavior and increasing moderate-intensity PA [57]. It has been indicated that prolonged sitting time was associated with lower health-related quality of life [58], and the reduction of sedentary behavior and the increase of PA bring about improved functional fitness in older adults [59]. However, a large number of older adults did not attain the recommended level of PA, and the proportion of inactive older people increases with age rapidly [60]. Active lifestyle might confer benefits on fitness more than PA habit does, therefore, PA should be encouraged in assisted living facilities as a routine to replace sedentary activities.

This study had a few limitations. (1) It was cross-sectional, which prevented it from demonstrating causal relationship between regular PA and physical functional performance. Further longitudinal studies are required to reveal the long-term effects of PA habits. (2) We did not standardize the exercises in mode and intensity. The self-reported information of PA habits might be biased. Quantitative exercise prescriptions are needed in further studies.

## 5. Conclusions

In conclusion, we found that regular PA was associated with muscle strength, flexibility, and better functional performances in both community-dwelling older adults and frail ones in day care centers. However, we found no significant associations between PA frequency and physical functional performance in healthy older adults, and it may be because their PA level is insufficient to show the effects. The results implied that currently physically independent older adults should maintain their effective amounts of PA to prevent frailty or disability. Our results also confirm the importance of regular PA as a routine to reduce sedentary time for the frail older adults in assisted living facilities.

## Figures and Tables

**Figure 1 ijerph-18-03561-f001:**
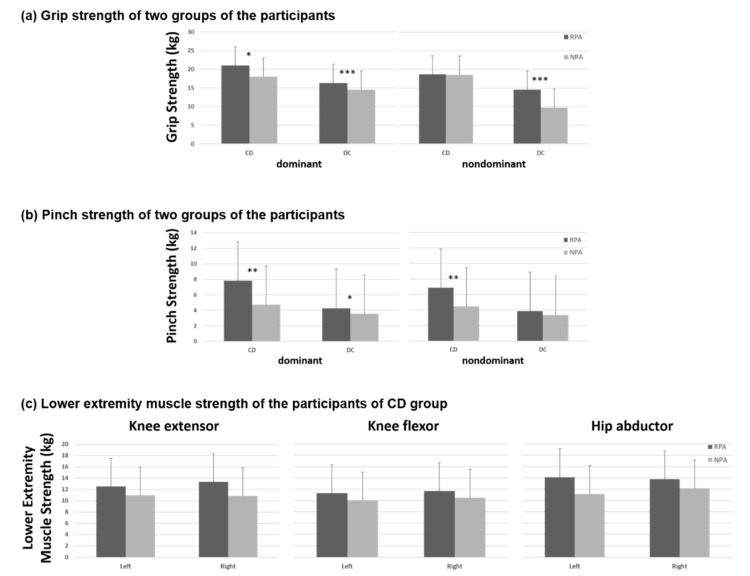
Muscle strength of (**a**) grip (**b**) pinch and (**c**) lower extremity of the participants; * *p* < 0.05; ** *p* < 0.05; *** *p* < 0.01.

**Figure 2 ijerph-18-03561-f002:**
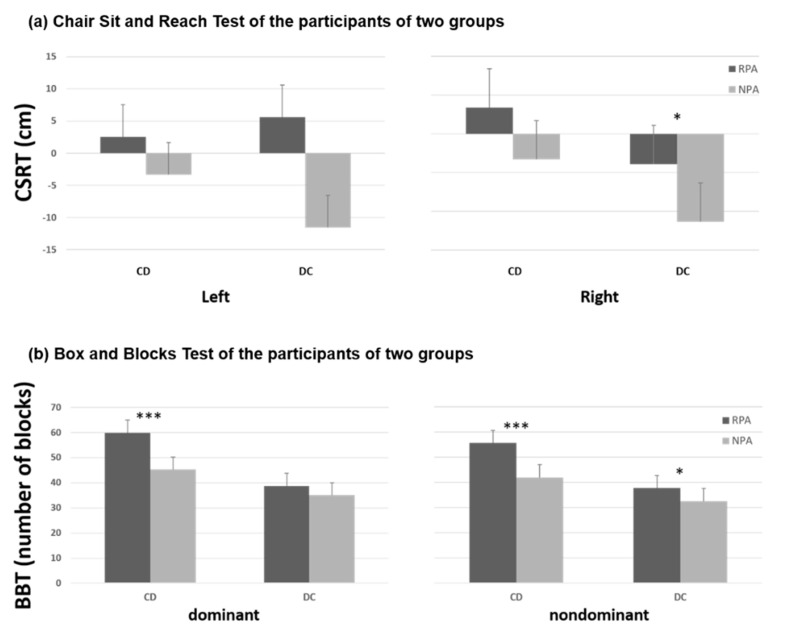
(**a**) Chair sit and reach test and (**b**) box and block test of the participants of two groups; * *p* < 0.05; *** *p* < 0.01.

**Figure 3 ijerph-18-03561-f003:**
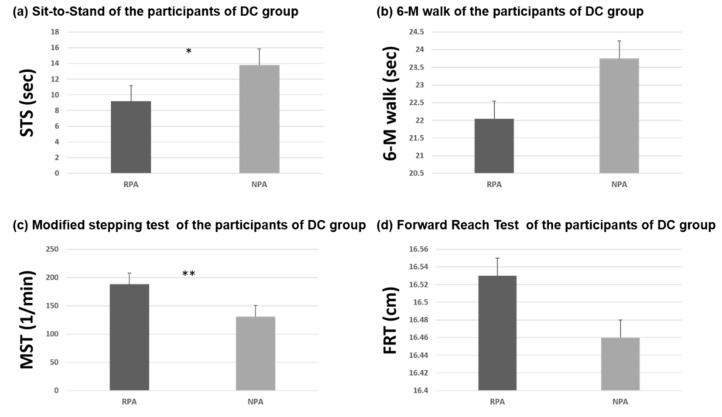
Functional test of (**a**) sit to stand test, (**b**) 6-m walking test, (**c**) modified stepping test and (**d**) forward reach test of the participants of DC group; * *p* < 0.05; ** *p* < 0.05.

**Figure 4 ijerph-18-03561-f004:**
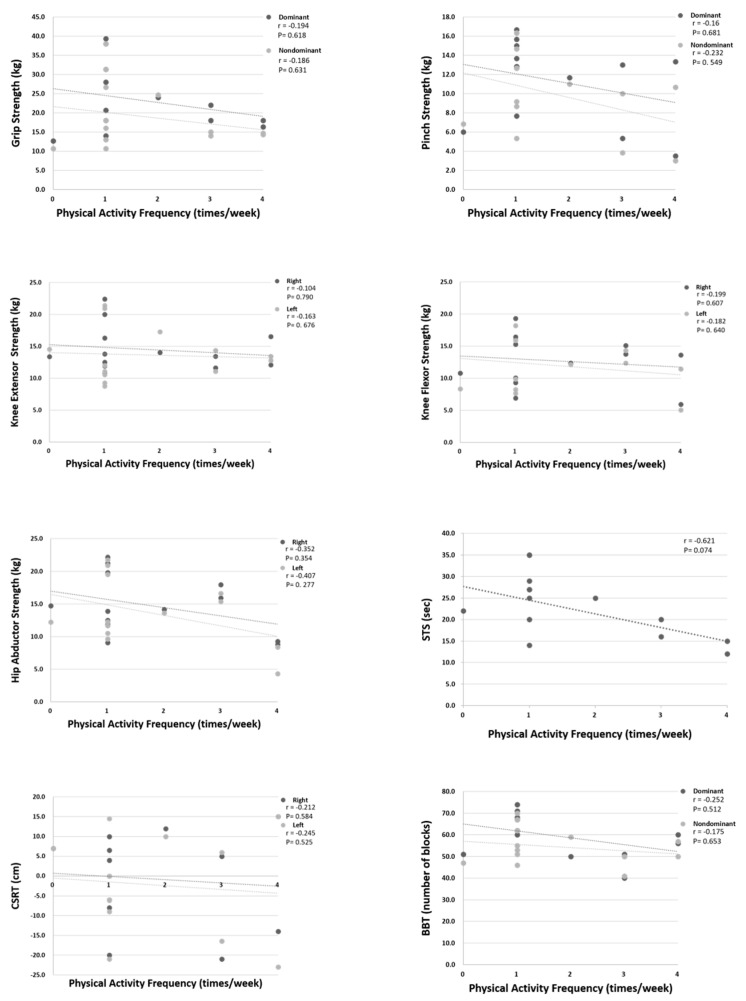
Correlations between exercise frequency and functional performances in the CD group.

**Table 1 ijerph-18-03561-t001:** Characteristics of the participants (average ± SD).

	Community-Dwelling (CD) Group		Day Care (DC) Center Group	
	RPA subgroup	NPA subgroup	RPA subgroup	NPA subgroup
Item	(*n* = 35)	(*n* = 40)	(*n* = 47)	(*n* = 41)
Age	74.4 ± 6.4 *	77.9 ± 7.3 *	79.8 ± 7.1	80.8 ± 6.8
Gender	Female: 26	Female: 22	Female: 35	Female: 32
	Male: 9	Male: 18	Male: 12	Male: 9
Height (cm)	154.99 ± 5.85	156.19 ± 8.91	151.62 ± 7.74	153.46 ± 9.1
Weight (kg)	57.23 ± 11.76	58.06 ± 9.24	54.70 ± 9.77 *	58.59 ± 8.77 *
BMI (kg/m^2^)	23.85 ± 4.93	24.23 ± 3.37	23.80 ± 3.81	25.04 ± 4.44

RPA: regular physical activity; NPA: non-physical activity; *n*: numbers of subjects. * *p* < 0.05.

**Table 2 ijerph-18-03561-t002:** Brief description of physical functional performance measures.

Item	Description	Measure
BBT	The subjects are instructed to move one block at a time from one compartment to the other one of the test box as fast as possible, but in one minute.	Number of blocks
CSRT	The subjects are seated on a chair with knee straight, then being instructed to place one hand on the other with tips of the middle fingers flush. The subjects reach slowly toward the toes by bending their trunks while exhaling. Distance between the fingers and the toes was measured. Average of two trials. *A negative distance means that the fingertips did not reach the toes, while a positive score means that the fingertips reached beyond the toes.	cm
Muscle strength	Measured by a hand dynamometer (Jamar^®^ Lafayette Instrument Company, Lafayette, IN, USA) for grip, a finger pinch gauge for pinch, and a digital dynamometer (MicroFET 3) for lower extremities. Average of three trials.	
grip	The subjects are seated, squeezing the hand dynamometer with their fingers as hard as they can without any body movement.	kg
pinch	The subjects are seated, squeezing the pinch gauge with their thumb and index fingers as hard as they can without any body movement.	kg
knee extensor	The subjects are seated, extending their knees against the resistance of the examiner applying on the anterior aspect of shanks proximal to their ankles.	kg
knee flexor	The subjects are prone, flexing their knees against the resistance of the examiner applying on the posterior aspect of shanks proximal to their ankles.	kg
hip abductor	The subjects are side lying, abducting their hips against the resistance of the examiner applying on the lateral aspect of their knee.	kg
STS	The subjects stand up and sit back from a straight-back chair three times as fast as possible with arm folded across the chest if possible. Time of three-time movements will be measured. Average of two trials.	sec
MST	The subjects are stepping alternatively as fast as possible while being seated for one minute. When one leg is raising, the foot should be completely off the ground. The score is the repetition number of stepping for one minute.	1/min
6-M walk	The subjects have to walk 6 m as fast as possible (with or without walking aid). Time of the movement will be recorded.	sec
FRT	The subjects are standing upright, stretching their arms forward, maintaining a fixed base of support while leaning forward is allowed. The distance between the front end of their fists before and after the movement will be measured. Average of two trials.	cm

**Table 3 ijerph-18-03561-t003:** Comparison of physical performances between the RPA and the NPA subgroups in CD group.

	CD-RPA	CD-NPA	*p*-Value
Item	(*n* = 35)	(*n* = 40)	
Grip (kg)			
dominant	21.05 ± 6.68	18.03 ± 5.25	0.031 *
nondominant	18.67 ± 6.18	18.57 ± 5.69	0.942
Pinch (kg)			
dominant	7.85 ± 3.89	4.74 ± 1.46	<0.001 ***
nondominant	6.86 ± 3.73	4.44 ± 1.47	0.001 **
CSRT (cm)			
right	3.40 ± 11.27	−3.32 ± 11.23	0.059
left	2.57 ± 11.58	−3.38 ± 11.32	0.101
BBT (number of blocks)			
dominant	59.9 ± 9.2	45.2 ± 11.9	<0.001 ***
nondominant	55.7 ± 7.6	42.0 ± 9.9	<0.001 ***
Knee extensor (kg)			
right	13.31 ± 3.46	10.82 ± 5.42	0.057
left	12.50 ± 3.45	10.94 ± 5.46	0.228
Knee flexor (kg)			
right	11.70 ± 3.01	10.51 ± 4.49	0.357
left	11.29 ± 2.99	10.04 ± 4.49	0.335
Hip abductor (kg)			
right	13.80 ± 3.42	12.16 ± 4.36	0.158
left	14.12 ± 7.26	11.16 ± 4.49	0.151

Data are expressed as means ± standard deviation. CD-RPA: regular physical activity subgroup in community-dwelling group; CD-NPA: non-physical activity subgroup in community-dwelling group; *n*: number of subjects; CSRT: chair sit-and-reach test; BBT: box and blocks test. *p* values based on the Mann–Whitney U test; * *p* < 0.05; ** *p* < 0.05; *** *p* < 0.01.

**Table 4 ijerph-18-03561-t004:** The comparison of physical performance between the RPA and the NPA subgroups in DC group.

	DC-RPA	DC-NPA	*p*-Value
Item	(*n* = 47)	(*n* = 41)	
Grip (kg)			
dominant	16.33 ± 7.81	10.71 ± 4.12	<0.001 ***
nondominant	14.55 ± 6.56	9.74 ± 6.07	0.001 ***
Pinch (kg)			
dominant	4.29 ± 1.49	3.57 ± 1.43	0.023 *
nondominant	3.86 ± 1.43	3.36 ± 2.72	0.286
CSRT (cm)			
right	−3.89 ± 8.53	−11.35 ± 12.95	0.031 *
left	5.61 ± 9.27	−11.55 ± 12.62	0.083
BBT (number of blocks)			
dominant	38.7 ± 10.1	35.0 ± 12.9	0.131
nondominant	37.8 ± 11.0	32.6 ± 11.8	0.039 *
STS (sec)	9.18 ± 5.96	13.83 ± 9.98	0.037 *
MST (1/min)	188.00 ± 62.14	130.54 ± 46.51	0.002 **
6-M walk (sec)	22.04 ± 20.66	23.75 ± 20.29	0.772
FRT (cm)	16.53 ± 6.26	16.46 ± 9.36	0.98

Data are expressed as means ± standard deviation. DC-RPA: regular physical activity subgroup in day care group; DC-NPA: non-physical activity subgroup in day care group; *n*: number of subjects; CSRT: chair sit-and-reach test; BBT: box and blocks test; STS: sit-to-stand test; MST: modified stepping test; 6-M walk: six-meter walking test; FRT: forward reach test. *p* values based on the Mann–Whitney U test; * *p* < 0.05; ** *p* < 0.05; *** *p* < 0.01.

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
