# Peer review of "The Comparisons of Physical Functional Performances between Older Adults with and without Regular Physical Activity in Two Different Living Settings"

_ijerph, 2021, doi:10.3390/ijerph18073561_

Round 1

Reviewer 1 Report

Thank you for the opportunity to review this paper.

The article is interesting, however a would recommend a few changes.

The title of the paper is: How does Regular Physical Activity Influence Older Adults in 2 Different Living Settings?

1.Line 83-85: “Each group was further classified into “Regular Exercise” (RE) and “Non-Exercise” (NE) subgroups. Regular exercise was defined as engaging in physical activities at least three times a week for more than 30 minutes each time.”

Physical activity level was based only on answer on this question?

No questionnaire was used to assess physical activity while the study is focused on PA.

  1. 163 participants were included – it is not clear what was the drop-out rate, what was the randomization method, how many participants were excluded and why?
  2. Ethical approval number is not written, please add this information.
  3. What do you mean by “against the resistance of examiner’ Was is subjective measure like Lovett test? How were kg of resistance evaluated?
  4. Results are insufficiently described.

Based on such results, my suggestion is to compare physical functions between different adults’ groups. To compare PA with physical functions, a validated tool is recommended.

Author Response

Point 1: Line 83-85: “Each group was further classified into “Regular Exercise” (RE) and “Non-Exercise” (NE) subgroups. Regular exercise was defined as engaging in physical activities at least three times a week for more than 30 minutes each time.” 

Response 1: Thanks for the comment. In the Community-Dwelling (CD) older adults group, face-to-face questionnaires were used to collect data of the exercise habit, including frequency, intensity, time, and type of the exercise. We classified the individuals who engages in dynamic activities at least three times a week for more than 30 minutes each time with the Regular Exercise (RE) subgroup. The others were classified as No-Exercise (NE)subgroups. In the elders attending senior day care (DC) centers, according to the activity schedules of the day care centers, the ones in the centers with routine gross motor activities rather than static activities (such as arts and crafts activities) at least three times a week for more than 30 minutes each time were classified as RE subgroup, while the others as NE subgroup.

Point 2: 163 participants were included – it is not clear what was the drop-out rate, what was the randomization method, how many participants were excluded and why?

Response 2: Thanks for the comment. This is a cross-sectional study. The information of physical functional performance measures in the participants was collected just for once during the same period. The data shown in this article means successfully-measured data from the 163 participants. The individuals with failed measurement of physical functional performance were excluded. The failure reasons include insufficient cognition and poor physical capacity that they could not follow the instructions from the examiners or afford the tasks. However, in the recruitment criteria, we have excluded the individuals with major musculoskeletal disease or other disorders affecting their ability to perform the exercises and our physical function tests. Therefore, few participants dropped out in the experiment. We classified the participant by their lifestyles and exercise habits instead of randomization method.

Point 3: Ethical approval number is not written, please add this information.

Response 3: Thank you for reminding. We have added the information of the ethical approval number (Line 98).

Point 4: What do you mean by “against the resistance of examiner’ Was is subjective measure like Lovett test? How were kg of resistance evaluated?

Response 4: Thanks for the comment. We used a digital dynamometer (MicroFET 3) to assess the participants’ muscle strength of the lower extremities by manual muscle testing (MMT). According to the method of MMT, the examiner put the dynamometer and applied resistance on the specified places of the participants’ segments, and the participants were instructed in specific initial positions and directions of the movement. The dynamometer can display the measured strength in the unit “kg”.

Point 5: Results are insufficiently described.

Response 5: Thanks for the comment. We have revised the contents in Results.

Based on such results, my suggestion is to compare physical functions between different adults’ groups. To compare PA with physical functions, a validated tool is recommended.

Reviewer 2 Report

Attached file

Author Response

Point 1: Introduction The paper is more interesting in relation to the effects of regular physical activity on physical functional performance in community-dwelling and day care centers older adults. The research aim is clearly defined. It would be necessary to expand and update the bibliographic review to justify the research problem.

Response 1: Thanks for the comment. We have revised the section of Introduction by expanding and updating more literature reviews to highlight the research problem.

Point 2: Methods Although the sample size is small, this data is recognized in the limitations.

Response 2: We appreciate the reviewer’s comment.

Point 3: Results Authors must perform the following task: add a graph with the results (tables 3, 4 and 5)

Response 3: Thanks for the comment. We have added 4 graphs with the results of tables 3, 4, and 5.

Point 4: Discussion The discussion section must be updated with the new bibliographic references. References Bibliographic references from the last two years must be included.

Response 4: Thanks for the comment. We have updated several references including those of last two years.

Reviewer 3 Report

The study aims to study the effect of regular physical activity on physical function performances in 163 older adults recruited from two different living contexts (supposedly) in rural Taiwan. The researchers employed questionnaires to assess physical activity habits, whereas a set of tests were applied to measure physical functional performance. Based on non-parametric Mann-Whitney U tests and correlation calculations, the authors claim statistically significant associations between regular physical activity and muscle strength, flexibility and functional performances in older adults, both in the group living in a community-dwelling and the group of frail older adults in day-care centres. No significant association was found between exercise frequency and physical functional performance.  

Main comments

  1. The background and rationale for the study are based on a very limited selection of literature, which is also less relevant to the actual subject of the study. For example, few references are made to literature on physical performance or to previous studies of the relationship between physical activity and the outcome variable, whereas 12 studies about the beneficial effect of physical activity on health are cited. Partly due to the limited amount of literature, the authors are unable to recognize the importance of other research, and for the same reason, they fail to identify the knowledge gap that their study is supposed to close. Moreover, because the text is difficult to read, the introduction, unfortunately, appears both unstructured and not very well prepared.
  2. Key concepts are neither properly defined nor operationalized. For example, it is not clear what the authors mean by regular physical activity and habitual physical (in)activity, and the important distinctions between physical activity and exercise have not been taken into account. In addition, the authors use the terms 'normal old adults' (line 40) and 'normal/healthy older adults' (lines 61, 98) without explaining the difference between these and other ‘not normal/unhealthy older adults’. Similar objections may also apply to the terms ‘functional fitness’, ‘physical fitness’, 'physical degeneration' and 'inactive or frail'; in any case, inconsistent use of terms emphasises an unnecessarily unclear presentation of the study's rationale. Of paramount importance, however, is that the study's two most important variables, namely "regular physical activity" and "physical functional performance", are neither defined nor described and explained in more detail.

I recommend that besides making use of proofreading services from native English speakers, the Introduction is revised with regard to previous literature and key concepts are explained and clarified.

  1. The method chapter should be improved on the following:

  1. The participants should be described in more detail, not least because the descriptions of the two living settings from which the participants are recruited can be interpreted differently across countries. It is not obvious what is meant by community dwellings or daycare centres, or the difference between them.
  2. As mentioned above, the variables should be defined and operationalized so that the reader knows exactly what the researchers have studied and how it has been measured. For example, the authors describe that frequency of physical activity was measured using a questionnaire, but there is a lack of information about how the question was asked and how it could be answered. Also, it is not clear what is meant by ‘dominant’ and ‘nondominant’ grip/pinch. Such information is crucial for any possible replication of the study. I would also prefer an accurate description of the variable cut-offs. For example, the authors describe that regular exercise was defined as engaging in physical activities at least three times a week for more than 30 min each time, nevertheless, it is not entirely clear what is defined as non-exercise. Besides, the somehow mixed use of the terms ‘physical activity’ and ‘exercise’ should be avoided as the latter is a subordinate of the first.
  3. The section about physical functional performance may appear unclear. A reference is made to the literature in line 98, without any literature being cited. Also, the section suffers from the inadequate clarification of concepts mentioned earlier. This section would be easier to understand if the reader has been introduced to the applied concepts beforehand, although an improvement and clarification of this section would enhance the text even more.
  4. In the 'statistical analysis’ section, the authors describe that they have used t-tests for analysing differences between groups. The tables, however, describe p-values ​​for Mann-Whitney U tests. Although it seems reasonable that the latter non-parametric tests were applied, there is an inconsistency between the description in the main text and the subtext of the tables. As a basis for the choice of statistical analyses, I would also prefer that the authors explain how they tested for normality.

  1. The manuscript suffers severe citing issues, for example:
    1. The paper lacks references (e.g. line 43, 44 and 45, and in line 99 where the authors refer to ‘the literature’ without citing any).
    2. None of the physical functional performance tests described is accompanied by citations.
    3. In the discussion part, lines 151-153, the authors refer to results apparently mentioned in the introduction while the articles that are referred to (23, 24 and 25) have not been mentioned in the introduction. It is not clear which previous results the authors actually intended to confirm.

  1. The aim of the study is to study the effect of physical activity; however, the research design does not allow for causality thus I recommend the aim being revised.

  1. I find the discussion difficult to read, and the lack of references to previous relevant research that I mentioned previously applies to this section as well. The results from the author’s study should be discussed and related to similar studies in order to indicate what is different and new in their paper. Although there are some studies mentioned, the discussion appears limited.

Minor comments:

  1. Consider revising the article’s title, which now reads: ‘How does Regular Physical Activity Influence Older Adults in Different Living Settings?’ Because regular physical activity could influence older adults in many different ways it seems that the outcome variable (physical function performance) is missing here.
  2. The abbreviations of the physical functional performance tests should be written in brackets after the names of the tests are spelt out (first time), and not the opposite (i.e. line 105)
  3. The asterisks indicating the level of significance should be swapped from behind the ‘item’ to behind the p-value in Table 3 and Table 4
  4. I would prefer that Pearson’s correlation coefficient is referred to as ‘Pearson’s r’, and not PCC.
  5. The order in which the test results are presented differs between Table 3 and Table 4. In order to make it easier comparing these tables, I would have preferred if the tests conducted in both groups were arranged in the same order in both tables.

I hope the authors find these comments helpful in improving the manuscript.

Author Response

The study aims to study the effect of regular physical activity on physical function performances in 163 older adults recruited from two different living contexts (supposedly) in rural Taiwan. The researchers employed questionnaires to assess physical activity habits, whereas a set of tests were applied to measure physical functional performance. Based on non-parametric Mann-Whitney U tests and correlation calculations, the authors claim statistically significant associations between regular physical activity and muscle strength, flexibility and functional performances in older adults, both in the group living in a community-dwelling and the group of frail older adults in day-care centres. No significant association was found between exercise frequency and physical functional performance.  

Main comments

Point 1: The background and rationale for the study are based on a very limited selection of literature, which is also less relevant to the actual subject of the study. For example, few references are made to literature on physical performance or to previous studies of the relationship between physical activity and the outcome variable, whereas 12 studies about the beneficial effect of physical activity on health are cited. Partly due to the limited amount of literature, the authors are unable to recognize the importance of other research, and for the same reason, they fail to identify the knowledge gap that their study is supposed to close. Moreover, because the text is difficult to read, the introduction, unfortunately, appears both unstructured and not very well prepared.

Response 1: We have revised the section of Introduction to connect physical activity and the outcome variables. We also expanded more literature reviews to elucidate the point of importance of this study.

Point 2: Key concepts are neither properly defined nor operationalized. For example, it is not clear what the authors mean by regular physical activity and habitual physical (in)activity, and the important distinctions between physical activity and exercise have not been taken into account. In addition, the authors use the terms 'normal old adults' (line 40) and 'normal/healthy older adults' (lines 61, 98) without explaining the difference between these and other ‘not normal/unhealthy older adults’. Similar objections may also apply to the terms ‘functional fitness’, ‘physical fitness’, 'physical degeneration' and 'inactive or frail'; in any case, inconsistent use of terms emphasises an unnecessarily unclear presentation of the study's rationale. Of paramount importance, however, is that the study's two most important variables, namely "regular physical activity" and "physical functional performance", are neither defined nor described and explained in more detail.

I recommend that besides making use of proofreading services from native English speakers, the Introduction is revised with regard to previous literature and key concepts are explained and clarified.

Response 2: Thanks for the comment. We have checked the similar and amphibolous terms. We added definition of the terms that may be confused and revised the similar terms to maintain consistency.

The manuscript has been proofread by a native English speaker before submission. We agree that the revised manuscript should be proofread again before upload. However, due to only 2 weeks allowed for major revision, we apologize for not having much time to send the revised manuscript to English proofreader after completing the revision. We would be grateful if we are allowed to make this manuscript proofread by English speaker later.

Point 3: The method chapter should be improved on the following:

  1. The participants should be described in more detail, not least because the descriptions of the two living settings from which the participants are recruited can be interpreted differently across countries. It is not obvious what is meant by community dwellings or daycare centres, or the difference between them.
  2. As mentioned above, the variables should be defined and operationalized so that the reader knows exactly what the researchers have studied and how it has been measured. For example, the authors describe that frequency of physical activity was measured using a questionnaire, but there is a lack of information about how the question was asked and how it could be answered. Also, it is not clear what is meant by ‘dominant’ and ‘nondominant’ grip/pinch. Such information is crucial for any possible replication of the study. I would also prefer an accurate description of the variable cut-offs. For example, the authors describe that regular exercise was defined as engaging in physical activities at least three times a week for more than 30 min each time, nevertheless, it is not entirely clear what is defined as non-exercise. Besides, the somehow mixed use of the terms ‘physical activity’ and ‘exercise’ should be avoided as the latter is a subordinate of the first.
  3. The section about physical functional performance may appear unclear. A reference is made to the literature in line 98, without any literature being cited. Also, the section suffers from the inadequate clarification of concepts mentioned earlier. This section would be easier to understand if the reader has been introduced to the applied concepts beforehand, although an improvement and clarification of this section would enhance the text even more.
  4. In the 'statistical analysis’ section, the authors describe that they have used t-tests for analysing differences between groups. The tables, however, describe p-values ​​for Mann-Whitney U tests. Although it seems reasonable that the latter non-parametric tests were applied, there is an inconsistency between the description in the main text and the subtext of the tables. As a basis for the choice of statistical analyses, I would also prefer that the authors explain how they tested for normality.

Response 3: Thanks for the comment.

  1. We have added more descriptions about the selection criteria, recruitment conditions and classifications of the participants in details.
  2. We have revised the descriptions of physical functional performance measures in details in Table 2, and provided the references for the measurements of the tests (Line 135 to 140). We also added the descriptions of questionnaire for exercise habits about the type and the questions in it (Line 104 to 108). Determinations of hand and leg dominances were explained in Line 141 to 143. We made it clearer in defining “regular exercise” and “non-exercise” subgroups and clarified the difference between “physical activity” and “exercise”.
  3. We have added the citations of the literatures about the differences of physical functional performances between healthy and frail older adults. We also revised the contents about physical functional performance to make it clearer.
  4. We have revised the mistake in writing in the 'statistical analysis’ section. Because samples in this study are small, we determined the differences between two subgroups of both CD and DC groups by using a nonparametric test, Mann-Whitney U test which is appropriate when distribution of the data population is unknown and for small samples.

Point 4: The manuscript suffers severe citing issues, for example:

    1. The paper lacks references (e.g. line 43, 44 and 45, and in line 99 where the authors refer to ‘the literature’ without citing any).
    2. None of the physical functional performance tests described is accompanied by citations.
    3. In the discussion part, lines 151-153, the authors refer to results apparently mentioned in the introduction while the articles that are referred to (23, 24 and 25) have not been mentioned in the introduction. It is not clear which previous results the authors actually intended to confirm.

Response 4: Thanks for the comment.

  1. We have added cited reference in Line 48-55 and Line 130.
  2. We have revised the descriptions of physical functional performance tests in in more detail and provided the references of the tests.
  3. We described the associations between PA and age-related declines in Introduction by adding the references cited in the first paragraph so that we could discuss the issue in the later section.

Point 5: The aim of the study is to study the effect of physical activity; however, the research design does not allow for causality thus I recommend the aim being revised.

Response 5: Thanks for the comment. We agree the aim of the study should be revised since this is a cross-sectional study and there is no activity intervention involved. We have revised the aim in the last paragraph of Introduction.

Point 6: I find the discussion difficult to read, and the lack of references to previous relevant research that I mentioned previously applies to this section as well. The results from the author’s study should be discussed and related to similar studies in order to indicate what is different and new in their paper. Although there are some studies mentioned, the discussion appears limited.

Response 6: Thanks for the comment. We have revised the section of Discussion.

Minor comments:

Point 7: Consider revising the article’s title, which now reads: ‘How does Regular Physical Activity Influence Older Adults in Different Living Settings?’ Because regular physical activity could influence older adults in many different ways it seems that the outcome variable (physical function performance) is missing here.

Response 7: Thanks for the comment. We have changed article’s title to “The Comparisons of Physical Functional Performances between the Older Adults with and without Regular Physical Activity in Two Different Living Settings”, which accords more with the contents of the article.

Point 8: The abbreviations of the physical functional performance tests should be written in brackets after the names of the tests are spelt out (first time), and not the opposite (i.e. line 105)

Response 8: Thanks for the comment. We have revised it.

Point 9: The asterisks indicating the level of significance should be swapped from behind the ‘item’ to behind the p-value in Table 3 and Table 4

Response 9: Thanks for the comment. We have revised it.

Point 10: I would prefer that Pearson’s correlation coefficient is referred to as ‘Pearson’s r’, and not PCC.

Response 10: Thanks for the comment. We have revised it.

Point 11: The order in which the test results are presented differs between Table 3 and Table 4. In order to make it easier comparing these tables, I would have preferred if the tests conducted in both groups were arranged in the same order in both tables.

Response 11: Thanks for the comment. We have revised it.

I hope the authors find these comments helpful in improving the manuscript.

Response to the reviewer: Thanks for the Reviewer’s thorough review and useful suggestions. The authors have revised the manuscript regarding your constructive comments.

Round 2

Reviewer 1 Report

  1. It is still not clear whether respondents were excluded and how many subjects were recruited as only the final number of participants is reported. Each resident was recruited? It must be clearly reported who many participants were included and how many excluded and why. Please specify it.
  2. Where and when was the study carried out?
  3. Figures from Discussion should be moved to Results.
  4. Line 276 – PA level WAS associated

Thank you

Author Response

Response: Thanks for the comment.

  • We are sorry that we didn’t make a clear explanation on the participant recruitment with misleading information. We used a public subject recruitment of healthy community-dwelling older adults via two local bureaus in Tainan City, Taiwan. Inclusion criteria was listed in the announcement, therefore, all the individuals who signed up to participate in the study were eligible. Similar subject recruitment was used in the older adults attending senior day care centers. Because several tests in this study involved in ambulatory, independent ability of movements such as sit-to-stand and walking was regarded as the primary inclusion criteria. In the senior day care centers, all the individuals who were willing to participate in the study (but not not every resident in centers) satisfied the essential ability to perform the physical functional performance tests. Few subjects (less than 5) suspended the data measurements due to sudden discomforts on the test day (such as cold or dizziness) that would interrupt their performances, but they were re-tested another day. All the participants signed up to the study completed the physical functional performance tests. That is the reason why we said that finally 163 participants were recruited with inclusion criteria for this study and they were successfully-measured. In order to clarify this, we revised some description about subject recruitment. Please see Line 100-101.

  • This study was carried out in Tainan City, Taiwan, in summer of 2017.

  • We have moved figures from Discussion to Results.

  • From the results of our study, no significant association was found between PA frequency (times per week) and physical functional performances. This may because their physical activity level is insufficient to show the effects on physical functional performances. Therefore, we said that currently physically independent older adults should maintain their effective amounts of PA if they intend to prevent physical frailty or disability.

Reviewer 3 Report

Meanwhile, I have read the revised manuscript, which has improved, not least in terms of a significantly enhanced literature basis in the introduction. However, there are still a few points that should be amended.

- Reference number 1 seems not to provide the definitions of either physical activity or exercise and should be corrected. 

-I still would have preferred information on the exact wording in the questions asked for assessing exercise (or do you mean physical activity) habits. 

-I still get confused by how the terms exercise and physical activity are used in the text, e.g. in lines 115-117, the sentence is not logic, and in line 165-170, the terms are used inconsistently and not in accordance with the sub-categories (regular exercise / non-exercise). Besides, the discussion opens with a sentence about physical exercise, while according to its described purpose (and title), the study deals with physical activity. 

-Lines 112-113: "The exercise habits of the DC group were determined by the daily schedule of activity provided by the care centres". Are you sure these activities could be interpreted as physical exercise and not physical activities? Would such an interpretation appear easier to accept if you provided an example of these activities?

- Figures 1-4 are almost impossible to interpret due to their size and should be enlarged to increase the readability.

Author Response

Response: Thanks for the comment.

  • Thank you for pointing out the error. We have revised it.
  • Activity carried by major muscles is what we concerned in this study, this is, physical activity, while exercise is a specific form of physical activity. Since many older adults would overestimate the activity level when they were asked about their activity habits. We asked the participants about their activity regularity by using a familiar term of “exercise habit” in the face-to-face questionnaire. We asked for details about the type, duration, frequency, and intensity of their exercise habits, and identify the self-reported information. If the exercise habits of static activities, less than 30 minute a bout, and less than three times a week were reported, they will not be recognized as regular.
  • The main concern in this study is the impact of “physical activity” among older adults instead of specific exercise with a structed form. To exclude the problem of term confusion, we decided to change the terms “RE” and “NE” for subgroups to “RPA” and “NPA”, which mean “Regular Physical Activity” and “Non-Physical Activity”. We also revised the term “exercise” to “physical activity” in the manuscript that express our thoughts and research design for the consistency in the text. However, we kept the terms “exercise” or “physical activity” used in cited article without changing, because some of them refer to certain specific exercise, e.g. resistance, aerobic or balance exercises.
  • Before the subject recruitment, we have invited the senior day care centers in advance. The activity schedules of the day care center were different from each other, that some centers focus on physical training (e.g. walking and Tai Chi), while the others focus on static activities (e.g. drawing and handicraft arts). We checked and inspected the schedule of each center, then classified the participants attending the center with physical activity schedule under specific conditions (of duration and frequency) as Regular Physical Activity (RPA) group. The others were classified as Non-Physical Activity (NPA) group.
  • We have enlarged the sizes of figures 1-4 to increase the readability.
